# Translation, Cultural Adaptation, and Psychometric Evaluation of the Brazilian Portuguese Version of the Quality in Psychiatric Care—Outpatients Instrument

**DOI:** 10.3390/healthcare11071001

**Published:** 2023-03-31

**Authors:** Lars-Olov Lundqvist, Naiara Gajo Silva, Sônia Barros, Agneta Schröder

**Affiliations:** 1University Health Care Research Center, Faculty of Medicine and Health, Örebro University, 702 81 Örebro, Sweden; 2School of Nursing, Federal University of Mato Grosso do Sul, Coxim 79070-900, Brazil; 3School of Nursing, University of São Paulo, São Paulo 05508-900, Brazil; 4Institute of Advanced Studies, University of São Paulo, São Paulo 05508-900, Brazil; 5Department of Health Sciences in Gjøvik, Faculty of Medicine and Health Sciences, Norwegian University of Science and Technology (NTNU), 7034 Gjøvik, Norway

**Keywords:** community mental health services, outpatient care, quality of health care, psychometric evaluation, validation study

## Abstract

Measuring the quality of care received by patients of mental health services is necessary to determine the effectiveness of prevention programs and mental health treatment. This study translated the original Swedish Quality in Psychiatric Care—Outpatient (QPC-OP) instrument to Brazilian Portuguese, adapted it to the context of Brazilian psychosocial care centers (CAPS), and evaluated its psychometric properties. The instrument was translated and back-translated by two independent professional translators. A seven-person expert group of professionals and 31 psychiatric outpatients verified the content validity of the Brazilian Portuguese QPC-OP, which then was completed by 253 outpatients from 16 CAPS in São Paulo, Brazil. Confirmatory factor analysis revealed adequate goodness of fit for the factor structure corresponding to the original Swedish version, except for the discharge dimension. Three additional items added in the Brazilian Portuguese QPC-OP formed a separate factor. The internal consistency of the entire scale was excellent but low in some dimensions. In conclusion, the translation and cultural adaptation of the Brazilian Portuguese QPC-OP was satisfactory, and the psychometric evaluation demonstrated that the concept of quality of mental health care is similarly understood in the Brazilian and Swedish cultural context. Thus, the Brazilian Portuguese QPC-OP is a useful instrument for assessing the quality of care in the Brazilian CAPS context and will be useful in quality assurance and in cross-cultural research addressing quality of mental health care from the patient’s perspective.

## 1. Introduction

The monitoring of the quality of care received by patients of mental health services is necessary to determine the effectiveness of prevention programs and mental health treatment, as well as to strengthen the argument for increasing resources [1]. In recent years, there has been a growing interest in mental health service research in order to increase the involvement of patients in the development of outcome measures that assess the quality and effectiveness of the service provided [2], as well as to invite patients to participate in the evaluative processes of the measures [3].

In addition, knowledge of patients’ experiences with health care based on their own perspectives could offer us an understanding of how patients perceive the treatment they receive [4]. From this perspective, scales developed from the patients’ viewpoints will reflect what patients perceive as important. Consequently, patients’ knowledge can complement that of researchers and staff, which potentially increases the face validity and relevance of quality-of-care measures [2].

Quality is a multifaceted concept. To understand quality, it is necessary to consider the discourse of the actors involved in the evaluation and to understand the actors’ perceptions of quality [5]. This may vary according to the perspective of, or the role of, the person who defines quality, such as patients, family, managers, and researchers [6], as well as the object of interest [7]. Consequently, if we want to measure patients’ perceptions of quality of care, it is important that our questions are based on the patient’s own definition of quality of care.

In the Brazilian mental health context, there is no instrument to measure quality in psychiatric care from the patient’s perspective. Most research in Brazil that evaluates psychosocial care centers (CAPS—from the Portuguese “Centros de Atenção Psicossocial”) focuses on the evaluation of the structure of the health services using Donabedian’s health care quality model [8]. According to this model, the evaluation of structure implies the evaluation of the physical and organizational properties of the setting in which care is provided, without necessarily having a focus on the participation of patients in evaluation processes.

Mental health care in Brazil introduces CAPS as the main community mental health service. CAPS offer treatment to people who are suffering from severe mental health disorders, including care by a psychiatrist and the use of medications in association with therapeutic groups and multi-professional care, thus being effective in preventing hospitalizations (less than 10% of patients had a psychiatric hospital admission after treatment at the CAPS started) [9]. CAPS follow the Brazilian Mental Health Policy by working towards promoting the civil rights of people with mental disorders and helping people with mental illness develop autonomy [10].

Nevertheless, no instrument measuring the patients’ perceived quality of care adapted to the Brazilian context has been developed. However, an instrument is available that includes the patients’ perspectives in the evaluation process: the Quality in Psychiatric Care—Outpatient instrument (QPC-OP) [11]. The QPC-OP was developed from interviews with patients and by means of factor analysis and has been adapted and validated in Spanish [12] and Chinese [13].

Thus, the QPC-OP is a potentially useful instrument for evaluating patients’ perceptions of quality of care in Brazilian CAPS. To provide the QPC-OP as a useful tool for assessing the quality of care in CAPS, it is necessary to first translate, then adapt and validate the translated version, and verify that the items in the original version are also capable of representing the concept of quality of care in the Brazilian CAPS context. To guide us in this process, we used Sousa and Rojjanasrirat’s [14] guidelines, which are based on a review of published recommendations for the cross-cultural validation of instruments.

This study is part of a larger research project designed to adapt the QPC-OP for use in different international settings, test the psychometric properties and equivalence of dimensions in the different language versions, and describe and compare the quality of outpatient psychiatric care across countries. The specific aim of this study was to translate the QPC-OP to Brazilian Portuguese, adapt it to the Brazilian CAPS context, and to evaluate its psychometric properties.

## 2. Materials and Methods

### 2.1. The QPC-OP Instrument

The QPC-OP [11] is a self-administered multidimensional questionnaire consisting of 30 items comprising eight different dimensions of outpatient quality of care. The eight dimensions are encounter (six items), participation–empowerment (three items), participation–information (five items), discharge (three items), support (four items), environment (three items), next of kin (two items), and accessibility (four items). Each item is scored on a four-point Likert-type scale, with a rating from totally disagree (1) to totally agree (4). For each item, respondents can also answer “not applicable”. The original Swedish version has excellent psychometric properties, according to a Cronbach’s alpha of 0.95, a comparative fit index of 1.00, and a root mean square of approximation of 0.036 [11].

The QPC has been adapted to different psychiatric services: the QPC-OP for outpatient care [11], the QPC for psychiatric inpatient care (QPC-IP) [15], the QPC for psychiatric forensic inpatient care (QPC-FIP) [16], the QPC for daily activities in community-based services for people with psychiatric disabilities [17], and the QPC for the quality of community housing support [18]. Each QPC instrument also has a staff version.

### 2.2. Translation and Cross-Cultural Adaptation Process

We used Sousa and Rojjanasrirat’s [14] guidelines in the translation, adaptation, and validation process of the QPC-OP. The guidelines comprise seven steps; however, in accordance with Sousa and Rojjanasrirat, the optional and rarely used sixth step including bilingual evaluation of the pre-final version was omitted because there were no CAPS patients who were Brazilian Portuguese and Swedish bilinguals. Therefore, the process comprised six steps that describe the translation process (four steps), the adaptation process (one step), and the psychometric validation process (one step).

Steps 1–4. Translation and back translation.

The original version of the QPC-OP [11] was independently translated by two authorized translators—one native in Brazilian Portuguese and one native in the Swedish language (Step 1). Then, the Brazilian and Swedish research teams performed a synthesis. The translators were consulted if questions were raised (Step 2). Two other authorized translators performed blind backward translation and the Brazilian and the Swedish research teams compared the back-translations with the original version and made changes if needed (Step 3). All changes were made by consensus. When the two research teams considered themselves satisfied with the translation, it was scrutinized by a Brazilian expert committee (Step 4). The committee consisted of seven people: two CAPS patients, two professionals, two mental health teachers, and one specialist in quality of health care. To verify the validity of the content, the expert committee evaluated the degree to which each element of the Brazilian Portuguese QPC-OP was relevant and representative of the “high quality” construct. Instructions, items, and response options were considered elements of the Brazilian Portuguese QPC-OP.

Step 5. Pilot testing.

Thirty-one CAPS patients participated in the pilot testing. They were asked to assess whether the Brazilian Portuguese QPC-OP seemed appropriate for measuring the quality of care in CAPS (face validity) and whether the items of the Brazilian Portuguese QPC-OP were sufficient to represent the concept of quality (content validity). The content validity index (CVI) [19] was calculated.

Step 6. Psychometric evaluation.

The study was conducted at 16 CAPS units in the city of São Paulo, Brazil. There were about 5000 patients at the CAPS during the study period. We randomly selected 791 patients on the “active patients list”. Among them, 481 passed the inclusion criteria of being at least 18 years old, having been in treatment for at least three months, and being cognitively able to respond to the Brazilian Portuguese QPC-OP questionnaire either orally or in writing. Of these, 381 agreed to participate in the survey and 253 completed the Brazilian Portuguese QPC-OP form, giving a response rate of 52.6%. The patients whose educational level was eight years or higher were invited to answer the Brazilian Portuguese QPC-OP form by themselves in writing. To not exclude patients with reading difficulties, participants whose educational level was below eight years could choose to be interviewed and answer the form orally. One hundred and one patients answered the Brazilian Portuguese QPC-OP orally.

### 2.3. Data Collection

A mental health nurse evaluated the patients’ orientation ability, thought process, and attention. The patients contacted received a brief presentation on the purpose of the research project orally and were then invited by telephone or in a face-to-face meeting to participate. Subsequently, they were informed in writing and patients who agreed to participate signed an informed consent form.

### 2.4. Statistical Analyses

Using the R software version 3.3.2 [20] with the “lavaan” package [21], we conducted a confirmatory factor analysis (CFA) with diagonally weighted least-squares estimation performed on asymptotic covariance matrices. First, we input missing data by multiple imputations using multivariate imputation by chained equations with predictive mean matching using the “mice” package in R [22]. To verify the model’s goodness of fit, we used the following indices: the root mean square error of approximation (RMSEA), the standardized root-mean-square residual (SRMR), the comparative fit index (CFI), and the Tucker–Lewis index (TLI). We considered the minimum thresholds of acceptability for model fit as *p* < 0.05, RMSEA < 0.080, CFI > 0.90, and TLI > 0.90 [23]. For SRMR, an ideal score is <0.08, but a value as high as 0.08 to 0.1 is deemed acceptable [24]. Internal consistency was measured by Cronbach’s alpha [25], with a criterion for adequate consistency of 0.70 [26].

### 2.5. Ethical Considerations

Steps 1–5 of the study were approved by the Research ethics committee (CEP) of the School of Nursing of the São Paulo University (EEUSP; No. 1,235,315) and the Municipal Secretary of Health (SMS) of São Paulo (No. 1,244,469). All national research ethical precepts were obeyed as per the ordinance Portaria 466/12. The participation of the members of the expert committee (Step 4) who responded to the pre-test was voluntary and took place after signing the Free and Informed Consent Form. The pre-test data collection of CAPS patients (Step 5) occurred after authorization from the managers.

Step 6 of the study was approved by the University of Sao Paulo Nursing School research ethics committee and the São Paulo Health Secretary Ethics Committee. Potential patients at CAPS were informed in writing and those who agreed to participate signed an informed consent form.

## 3. Results

Steps 1–4. The translation–back-translation process.

There were a few discrepancies between the original instrument and the back-translation. Discrepancies were found in two items (17 and 19), which were discussed and resolved by the Brazilian and Swedish research groups. The Swedish “söka sysselsättning” in item 17 was first translated to “conseguir uma ocupação”, which in Portuguese is limited to formal work similar to the English “employment”. The item was revised to also include study and informal work, which is closer to the meaning of the Swedish “sysselsättning”. The Swedish “min omgivning” in item 19 was translated to “ambiente”. In Portuguese, this means “everything that surrounds or involves living beings and/or things”, which is not very specific and does not reflect the more specific meaning in Swedish of the item in this context: “people and objects around me”. Therefore, “ambiente” was changed to “pessoas e objetos ao meu redor”.

Thereafter, the translation was scrutinized by the expert committee, who suggested some further changes considering that the literal correspondence of a term does not necessarily imply similar interpretations in different cultures and that the degree of formality of the language can vary from country to country. Following the committee’s suggestions, the Portuguese translation “ambulatório” of the Swedish word “öppenvårdsmottagning”, equivalent to “outpatient clinic” in English, was replaced by “CAPS” in order to better correspond to the Brazilian context. Similarly, the Portuguese translation “sala de espera” of the Swedish word “väntrum”, equivalent to “waiting room” in English, was replaced by “nos espaços que compartilhamos” which is similar to “spaces we share” in English. Moreover, following the committee’s suggestions, the Swedish expression “behandlare/kontaktperson” equivalent to “health professional/contact person” in English was replaced by one expression, “profissional de referência”. The Portuguese translation “transtornos psíquicos” of the Swedish term “psykiska besvär”, equivalent to “psychological problems” in English, was replaced by “doença mental”, equivalent to “mental disorder”. Finally, to adjust to the CAPS context, the language was changed to a more colloquial style by modifying the more formal past tense of the Swedish version to the present tense in the Brazilian version.

Considering that the Brazilian Mental Health Policy aims to promote citizenship, and this subject is not included in the original QPC-OP, the expert committee suggested the inclusion of three new items addressing this issue in the Brazilian Portuguese QPC-OP to increase its usefulness in the Brazilian context. The new items were added as a “patients’ rights” dimension in the Brazilian Portuguese QPC-OP and were added at the end of the questionnaire to ensure that the original structure was affected as little as possible. The pre-version of the Brazilian Portuguese QPC-OP thus consists of 33 items.

Step 5. Pilot testing of the pre-version.

Among the thirty-one CAPS patients assessing the pre-version, seven indicated that they had some difficulty in understanding some of the items. The items that were most difficult to understand were item 19 (support dimension), pointed out by six participants; item 13, pointed out by three participants; and item 1, pointed out by two participants. When the questions were read aloud by a researcher, the number of participants with doubts was reduced to four, one, and none, respectively. Thus, self-rating instruments may represent a challenge for some CAPS patients to understand, indicating the need for an option to complete the questionnaire orally.

After adjustments were made, the version used in the pre-test was properly understood and considered relevant by the CAPS patients. The overall CVI-S was 97% and the degree of agreement (i.e., “totally adequate” and “adequate, but requiring minor revisions”) was equal to or greater than 86%, which is considered excellent [19].

Step 6. Full psychometric testing of the Brazilian Portuguese QPC-OP.

The participants’ sociodemographic and clinical characteristics are presented in Table 1. The typical patient was a white male who had no occupation, a low educational level, and no source of income.

We conducted a CFA of the original QPC-OP 30-item, 7-dimension model (Model 1). This model exhibited *χ*^2^ = 645.080, *df* = 377 (*p* < 0.001), CFI = 0.965, TLI = 0.960, RMSEA = 0.053, and SRMR = 0.097. Despite the favorable results, the model presented a problem due to a not positive definite covariance matrix of latent variables. The source of the problem was identified as the discharge dimension, which presented standardized coefficients larger than 1. In addition, two items of the discharge dimension had a factor loading smaller than 0.40. Therefore, we excluded the discharge dimension from further analyses.

Then, we analyzed Model 2, which comprised the remaining 27 items from the original QPC-OP, with the three items from the discharge dimension excluded from the model. The CFA of Model 2 showed a minor increase in goodness of fit, *χ*^2^ = 491.632, *df* = 303 (*p* < 0.01), CFI = 0.971, TLI = 0.966, RMSEA = 0.050, and SRMR = 0.093.

Because Model 2 showed adequate goodness-of-fit indices, we added the three Brazilian-specific items as a new dimension to the original seven QPC-OP dimensions. This model, Model 3, thus included 30 items in eight dimensions. The CFA of this model showed *χ*^2^ = 629.397, *df* = 435 (*p* < 0.01), CFI = 0.965, TLI = 0.960, RMSEA = 0.052, and SRMR = 0.097. As shown by the CFA, the Brazilian-specific items concerning the patients’ rights formed an independent dimension that expressed aspects of quality of care not covered by the original QPC-OP. As a result, the 30-item 8-dimension QPC-OP version expressed in Model 3 was deemed acceptable and potentially more useful for assessment within the Brazilian context. Thus, Model 3 was considered the final model. Summary statistics of Model 3, given in Table 2, show that the encounter dimension achieved the highest ratings, while the participation–information dimension performance had the lowest ratings. The mean of the entire instrument was 3.30; the highest mean was found for the next-of-kin dimension and the lowest was found for the patients’ rights dimension.

Model 3 showed adequate internal consistency for the full questionnaire, as well as in the encounter, support, and environment dimensions, but not in any of the other dimensions. As seen in Table 3, 3 (11%) of the correlation coefficients among the Brazilian Portuguese QPC-OP dimensions were weak (*r* < 0.30), 7 (25%) were moderate (*r* = 0.30–0.49), and 18 (64%) were strong (*r* ≥ 0.50) according to Cohen’s criteria [27].

## 4. Discussion

The main results of the study showed that the Brazilian Portuguese QPC-OP had excellent content validity and adequate psychometric properties. The semantic equivalence of the items was maintained, but some items had to be changed to fit the Brazilian context and ease readability, such as choosing a more colloquial language and present tense.

The Brazilian Portuguese QPC-OP underwent some modifications in order to adapt it to the Brazilian context. The changes were justified considering education level and socioeconomic differences between the populations of the two countries and some specificities of treatment in the CAPS. For instance, many patients in Sweden have completed higher education [28], while few in the Brazilian sample had completed primary education. Therefore, the wording in the Brazilian version was changed to a more colloquial language with shorter sentences compared to the original version, yet keeping a semantic equivalence and ensuring that the syntax and grammar of the items in the original version were maintained as far as possible in the translation.

Following the committee’s suggestions, three Brazilian-specific “patients’ rights” items were added. These items were considered important since they give the users the possibility to exercise their rights to express their opinion on the Brazilian mental health policy and allow the measurement of the quality of care specific to the context of psychosocial care at Brazilian CAPS, increasing the validity of the Brazilian Portuguese QPC-OP. The concept of discharge was discussed and questioned by the CAPS teams because discharging CAPS patients does not occur systematically. Despite this, we chose to keep the discharge items until the construct validity was assessed through CFA.

The assessments by the committee and the CAPS patients indicated the face and content validity of the pre-version of the Brazilian Portuguese QPC-OP, and in the final step of the process, we could determine that the model proposed in the original Swedish version of QPC-OP [11] was, to a large extent, applicable to the Brazilian context and that the patients’ rights items efficiently added to the original structure. Hence, the CFA showed that the Brazilian Portuguese QPC-OP replicated the factor structure of the original version, except for the items that constituted the discharge dimension. The goodness-of-fit indices showed that the data fitted Model 2 (without the discharge items) very well and that Model 3 (including the patients’ rights items) achieved a somewhat better fit, thus both models were deemed acceptable. Model 3, including the items concerning patients’ rights as suggested by the expert committee, showed adequate fit and by adding this dimension we potentially increased the validity and usability of the Brazilian Portuguese QPC-OP in the Brazilian context.

Regarding the discharge items, they may be less relevant to the CAPS situation, confirming the doubts expressed by the committee and the CAPS patients in Steps 4 and 5. CAPS are based on long-term contact with patients with mental disorders. Thus, most patients are enrolled in CAPS for a long time, in different treatment modalities that are more or less intensive. This means that few patients permanently terminate contact with the service. From this perspective, the discharge items in the Brazilian Portuguese QPC-OP may appear to be hypothetical and therefore less relevant to the patient’s situation, thus resulting in a poor model fit in the CFA.

All loadings were significant, showing that the items reflected the intended factor, and all but one, item 14 in the participation–information dimension, were above 0.40. Although the internal consistency reliability analysis using Cronbach’s alpha presented a favorable result for the full questionnaire and the encounter dimension, the remaining dimensions only approached adequate levels of internal consistency. The lowest alpha was observed for the next-of-kin dimension consisting of only two items, which is probably due to Cronbach’s alpha’s well-known sensitivity to the number of items. Thus, the performance of some Brazilian Portuguese QPC-OP dimensions may have been hampered by the small number of items that make up the dimension. The most important analysis in psychometric validation is validity, and low homogeneity is less worrisome if the factor loading is high [29].

Regarding the patients’ ratings of quality of care, the results showed that the mean rating for the entire instrument was 3.30 (the center of the scale is 2.5). This suggests that the patients perceived the quality of care as good. The highest mean rating was in the next-of-kin dimension. Thus, the results show that patients and their families participate in the planning and implementation of patients’ care plans as desired and as recommended by the Brazilian Mental Health Policy. The Brazilian Mental Health Policy also aims to promote citizenship and fight stigma and prejudice. Despite this, the lowest mean ratings were found for the patients’ rights dimension. This suggests that greater efforts are needed to implement the mental health policy at CAPS. However, we need to consider the complexity of implementing the policy to promote citizenship of people with mental disorders because the exclusion and loss of their rights was based on multifaceted historical processes. The Brazilian Mental Health Policy aims to change the social patterns left by these processes, which may still present barriers to building the patients’ citizenship [30,31].

Although the results affirm the reliability and validity of the Brazilian Portuguese QPC-OP, the present study has some limitations. First, the data collection was limited to São Paulo City. Because Brazil is a large country with several cultural and economic differences by region, the results of this study may not be fully generalizable to CAPS in rural areas of Brazil. Second, our sample size followed the literature recommendation, that is, 5–10 people per item [23]. However, it is probable that a larger sample size might have positively influenced the internal consistency reliability coefficients and SRMR results. Third, this was a cross-sectional study; thus, we cannot evaluate the re-test reliability of the Brazilian Portuguese QPC-OP. Given these issues, it is important to carry out new studies that overcome the limitations of this study.

## 5. Conclusions

The translation and cultural adaptation of the original QPC-OP was satisfactory and the psychometric evaluation demonstrated that the concept of quality of mental health care is, to a large extent, similarly understood in the Brazilian and Swedish cultural context. Thus, the Brazilian Portuguese QPC-OP was found to be a valid and reliable instrument for evaluating quality of care from the patient’s perspective in the Brazilian CAPS context. From a clinical perspective, and to the best of our knowledge, this is the first validated instrument in Brazilian Portuguese for measuring quality of care from the patient’s perspective in Brazilian CAPS. Hence, the Brazilian Portuguese QPC-OP instrument will potentially be useful in clinical practice in order to assess quality assurance and to guide and monitor quality improvement in CAPS.

## Figures and Tables

**Table 1 healthcare-11-01001-t001:** Sociodemographic and clinical characteristics of participants in Step 6, full psychometric testing (*n* = 253).

Variable	*n*	%
Sex ^#^		
Female	117	46.4
Male	135	53.6
Age group		
18–29	36	14.4
30–39	64	25.6
40–49	72	28.8
50–59	56	22.4
60–69	22	8.8
Race ^##^		
Asian	8	3.1
Black	26	10.3
Brazilian Indian	3	1.2
Mixed race	102	40.3
White	114	45.1
Occupation		
Student	14	5.5
Formal occupation	26	10.3
Informal occupation	36	14.2
No occupation	177	70.0
Education ^#^		
0–4 years of schooling	13	5.2
5–8 years of schooling	109	43.2
9–11 years of schooling	97	38.5
>11 years of schooling	33	13.1
Income ^#^		
Has a source of income	159	63.1
Has no source of income	93	36.9

^#^ One missing value; ^##^ Racial classification according to the Brazilian Institute of Geography and Statistics (IBGE) [26].

**Table 2 healthcare-11-01001-t002:** Summary statistics of confirmatory factor analysis of the Brazilian Portuguese QPC–OP for patients of Brazilian outpatient services (*n* = 253).

Brazilian Portuguese QPC-OP Items by Dimension	Loading *	Cronbach’s α	Mean	SD
Full Brazilian Portuguese QPC-OP		0.87	3.30	
1. Encounter		0.77	3.59	
11. Shows empathy	0.76		3.69	0.69
12. Cares if I get angry	0.60		3.41	1.00
15. Respects me	0.86		3.79	0.58
18. Shows understanding	0.77		3.45	0.95
20. Has time to listen	0.73		3.52	0.89
25. Cares about my care	0.80		3.66	0.75
2. Participation–empowerment		0.58	3.13	
1. Influence over my care	0.50		3.18	1.08
5. My view of the right care is respected	0.86		3.36	1.02
6. Take part in decision making about my care	0.57		2.83	1.18
3. Participation–information		0.57	3.09	
13. Benefit drawn from earlier experience of treatment	0.56		3.46	0.96
14. Recognize signs of deterioration	0.25		3.30	1.08
27. Given information in a way that can be understood	0.55		2.91	1.20
29. Knowledge about mental troubles	0.67		3.15	1.12
30. Information about treatment alternatives	0.66		2.64	1.27
4. Support		0.64	3.12	
19. Stops me from hurting others	0.66		3.21	1.08
22. Stops me from hurting myself	0.61		3.22	1.10
23. Nothing shameful about having mental troubles	0.87		3.04	1.29
24. Shame and guilt must not get in the way	0.80		2.98	1.33
5. Environment		0.64	3.41	
2. High level of security at clinic	0.70		3.57	0.80
4. Feel secure with fellow patients	0.76		3.33	1.04
9. Not disturbed by fellow patients	0.74		3.33	1.01
6. Next of kin		0.41	3.68	
10. Next of kin invited to take part	0.51		3.57	0.89
28. Respects my next of kin	0.87		3.79	0.59
7. Accessibility		0.55	3.14	
3. Easy to meet the contact person	0.74		3.27	1.04
7. Easy to get an appointment	0.63		3.29	1.07
16. Easy to reach the clinic by phone	0.74		3.65	0.75
26. Easy to meet the doctor	0.51		2.36	1.29
8. Patient rights		0.51	3.06	
31. Receive information about “territory”	0.66		3.15	1.14
32. Feel comfortable to participate actively in the CAPS meeting	0.45		3.19	1.18
33. Received information about my rights	0.74		2.83	1.23

* All loadings *p* < 0.05.

**Table 3 healthcare-11-01001-t003:** Correlation coefficients of the Brazilian Portuguese QPC-OP dimensions (*n* = 253) ^#^.

Brazilian Portuguese QPC-OP Dimensions	1	2	3	4	5	6	7	8
1. Encounter	1.00							
2. Participation–empowerment	0.77	1.00						
3. Participation–information	0.72	0.78	1.00					
4. Support	0.51	0.49	0.67	1.00				
5. Environment	0.66	0.58	0.38	0.29	1.00			
6. Next of kin	0.64	0.59	0.48	0.40	0.87	1.00		
7. Accessibility	0.74	0.65	0.75	0.35	0.61	0.68	1.00	
8. Patient rights	0.38	0.67	0.73	0.36	0.13	0.29	0.59	1.00

^#^ All correlation coefficients are significant at *p* < 0.05.

## Data Availability

The data presented in this study are available on request from the second author, Naiara Gajo Silva, due to privacy reasons.

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
