# Peer review of "Translation, Cultural Adaptation, and Psychometric Evaluation of the Brazilian Portuguese Version of the Quality in Psychiatric Care—Outpatients Instrument"

_healthcare, 2023, doi:10.3390/healthcare11071001_

Round 1
Reviewer 1 Report
I congratulate the authors for the work carried out, which I consider to be of great relevance.
The procedure used in the translation, cultural adaptation, and psychometric evaluation of the QPC-OP into Brazilian Portuguese seems adequate and correctly explained to me. There are, however, some aspects to which I request the authors' attention:
1. There should be a standardisation of the way they refer to the instrument translated and validated. In the title, it is referred as the "Brazilian Portuguese version of the Quality in Psychiatric Care - Outpatients instrument" which is, in fact, the most adequate; however, throughout the text, it is verified that this designation is not always used, appearing many times as "Brazilian version" c.f. line 23, 209, 211 and 225, for example, or only QPC-OP when refering to the translated version, c.f. line 27. This situation is more notorious during the "Discussion" in which different designations are used for the same instrument, c.f. lines 273 - "Brazilian Portuguese QPC-OP; line 278 and 299- "Brazilian version of QPC-OP and in line 339 - "Brazilian QPC-OP".
2. I have some doubts in the explanation given for the sample selection. From reading the text presented between lines 131 and 142, the following questions remain:
a) How did you arrive at the number 791 for the sample size?
b) Of the 481 participants who passed the inclusion criteria, 381 agreed to participate, and of these, 253 completed the QPC-OP. Why do you consider that the response rate was 52.6% (253 out of 481) when if in fact there are responses from 253 out of 381, who are the ones participating in the study? should it not be a response rate of 66.4%?
c) The written response option was given to participants with a higher educational level. However, contact was made by telephone or in person. How was the written response obtained from those contacted via telephone? and of those with more than 8 years of educational level, how many chose to respond orally?
Author Response
An account of the revision of manuscript ID ijerph-2291223 entitled “Translation, cultural adaptation and psychometric evaluation of the Brazilian Portuguese version of the Quality in Psychiatric Care – Outpatients instrument”
Dear Lydia Zeng and anonymous reviewer,
Thank you for reviewing our manuscript. We have given full consideration to the comments made by the reviewer. The new version of the manuscript presents a complete revision in accordance with these comments. All changes are marked using the “Track Changes” function in MS Word. A description of how we have addressed the comments is detailed below.
It is our hope that you and the reviewer will regard the revised version as a considerable improvement.
We look forward to hearing from you.
Reviewer 1:
R1.1
There should be a standardisation of the way they refer to the instrument translated and validated. In the title, it is referred as the "Brazilian Portuguese version of the Quality in Psychiatric Care - Outpatients instrument" which is, in fact, the most adequate; however, throughout the text, it is verified that this designation is not always used, appearing many times as "Brazilian version" c.f. line 23, 209, 211 and 225, for example, or only QPC-OP when refering to the translated version, c.f. line 27. This situation is more notorious during the "Discussion" in which different designations are used for the same instrument, c.f. lines 273 - "Brazilian Portuguese QPC-OP; line 278 and 299- "Brazilian version of QPC-OP and in line 339 - "Brazilian QPCOP".
Authors’ response: Thank you very much for this comment. We fully agree that our treatment of the term QPC-OP in different ways creates ambiguity. We have therefore chosen to refer to the Brazilian version as "The Brazilian Portuguese QPC-OP" throughout the manuscript.
R1.2
I have some doubts in the explanation given for the sample selection. From reading the text presented between lines 131 and 142, the following questions remain:
- a) How did you arrive at the number 791 for the sample size?
Authors’ response: Thank you very much for your comment. We understand that a planned sample size of 791 may raise questions. The reason is that we wanted:
1) a sample that reflected the proportion of CAPS III and CAPS II patients. Thus, the number of patients in CAPS III should be 26% larger than CAPS II and
2) the sample size should meet the psychometric recommendation of minimum 100 participants per group (i.e., 100 CAPS II and 126 CAPS III; total n=226), and
3) the sample should take into account dropouts due to outdated patient addresses or not filled in CAPS forms. Based on our experience from previous studies we decided to multiply our desired sample by 3.5.
Hence, the sample size was determined by 226 x 3.5 resulting in a sample size of n=791.
- b) Of the 481 participants who passed the inclusion criteria, 381 agreed to participate, and of these, 253 completed the QPC-OP. Why do you consider that the response rate was 52.6% (253 out of 481) when if in fact there are responses from 253 out of 381, who are the ones participating in the study? should it not be a response rate of 66.4%?
Authors’ response: We followed recommended methods for calculating the response rate, which states that the number of respondents should be divided by the number of invitees, thus we arrive at a response rate of 52.6% (253 out of 481).
- c) The written response option was given to participants with a higher educational level. However, contact was made by telephone or in person. How was the written response obtained from those contacted via telephone? and of those with more than 8 years of educational level, how many chose to respond orally?
Authors’ response: Thank you for the comment, however, we suspect that there might be some misunderstanding. In the text we state that: "The, patients contacted received a brief presentation on the purpose of the research project orally and were then invited by telephone or in a face-to-face meeting to participate. Subsequently, they were informed in writing and patients who agreed to participate signed an informed consent form.” Thus, no participant answered the questionnaire over the phone. They were only invited to participate in the survey by telephone or/and in person.
Reviewer 2 Report
Thank you for submitting your paper regarding the ‘Translation, cultural adaptation and psychometric evaluation of the Brazilian Portuguese version of the Quality in Psychiatric Care – Outpatients instrument’. The introduction, background, and rationale for your study are all clearly articulated. The statistical methods and data analyses summarized are suitable and sound.
I have the following suggestions for your consideration.
Line 98, provide the figure of reliability and validity to support your argument ‘the excellent psychometric properties’.
Line 108, If there are CAPs patients who were Brazilian Portuguese and Swedish bilinguals in future.
Line121, please be specific on the member of committee.
Line 165, please cover the name of university and the project number.
Line 224, add reference to support your conclusion ‘86% which is excellent’.
Line 227, what is the meaning of typical? Avoid using ‘White’, and ‘Black’ in your manuscript.
Line 245, provide a table ‘selected fit indexes for three models…’
Author Response
An account of the revision of manuscript ID ijerph-2291223 entitled “Translation, cultural adaptation and psychometric evaluation of the Brazilian Portuguese version of the Quality in Psychiatric Care – Outpatients instrument”
Dear Lydia Zeng and anonymous reviewer,
Thank you for reviewing our manuscript. We have given full consideration to the comments made by the reviewer. The new version of the manuscript presents a complete revision in accordance with these comments. All changes are marked using the “Track Changes” function in MS Word. A description of how we have addressed the comments is detailed below.
It is our hope that you and the reviewer will regard the revised version as a considerable improvement.
We look forward to hearing from you.
Reviewer 2:
Authors’ response: Thank you very much for valuable comments. We have responded to your comments in the following way:
R2.1
Line 98, provide the figure of reliability and validity to support your argument ‘the excellent psychometric properties’.
Authors’ response: We have added figures of reliability and validity described in the referred study.
R2.2
Line 108, If there are CAPs patients who were Brazilian Portuguese and Swedish bilinguals in future.
Authors’ response: Unfortunately there were no Brazilian Portuguese and Swedish bilinguals among the CAPS patients. As stated by Sousa and Rojjanasrirat this step is optional. However, we agree that in future evaluations a bilingual evaluation should be performed if possible.
R2.3
Line121, please be specific on the member of committee.
Authors’ response: The text states: “The committee consisted of seven people: two CAPS patients, two professionals, two mental health teachers and one specialist in quality of health care”, which may be regarded as sufficiently specific.
R2.4
Line 165, please cover the name of university and the project number.
Authors’ response: Thank you for the comment, however, according to the journal instructions, one should openly report funders and who has approved the research ethics application.
R2.5
Line 224, add reference to support your conclusion ‘86% which is excellent’.
Authors’ response: Thank you for this observant comment. A reference to support the claim has been added.
R2.6
Line 227, what is the meaning of typical?
Authors’ response: By “typical” we mean the most common characteristic among the participants.
R2.7
Avoid using ‘White’, and ‘Black’ in your manuscript.
Authors’ response: Thank you for this important question. We understand the desire to avoid concepts that may be perceived as derogatory or exclusionary. In the research group, we have discussed this intensively. Especially since descriptions of individuals based on skin color are considered inappropriate from a Swedish perspective. But this study is being carried out in Brazil and we then choose to describe the participants in accordance with the Brazilian official nomenclature. In Brazil it is recommended to include a racial perspective in health surveys to identify (racial) health inequalities. The variable race/color (white, black, yellow, brown and indigenous) is self-declared, and emerged as a solution to the diversity of ethnic-racial identities declared in the Censuses of Brazil. To clarify this, we have included a reference to the Brazilian Institute of Geography and Statistics in Table 1. We have also studied recent published articles in IJERPH that relate to Brazilian conditions and in these articles the above race/color terms are used.
R2.8
Line 245, provide a table ‘selected fit indexes for three models…’
Authors’ response: Thank you very much for valuable suggestion. However, we have chosen to report goodness-of-fit indices in body text to keep the number of tables in the manuscript at a reasonable level to facilitate readability.